# Reporting delays of chikungunya cases during the 2017 outbreak in Lazio region, Italy

**Mattia Manica**[1,2,3]*, **Giovanni Marini**[2,3], **Angelo Solimini**[4], **Giorgio Guzzetta**[1,3], **Piero Poletti**[1,3], **Paola Scognamiglio**[5], **Chiara Virgillito**[4], **Alessandra della Torre**[4], **Stefano Merler**[1,3], **Roberto Rosà**[2,6], **Francesco Vairo**[5], **Beniamino Caputo**[4]

**1** Center for Health Emergencies, Bruno Kessler Foundation, Trento, Italy, **2** Research and Innovation Centre, Fondazione Edmund Mach, San Michele all'Adige (TN), Italy, **3** Epilab-JRU, FEM-FBK Joint Research Unit, Trento, Italy, **4** Department of Public Health and Infectious Diseases, Università di Roma SAPIENZA, Rome, Italy, **5** Regional Service for Surveillance and Control of Infectious Diseases (SERESMI) —Lazio Region, National Institute for Infectious Diseases "Lazzaro Spallanzani"; IRCCS, Rome, Italy, **6** Center Agriculture Food Environment, University of Trento, San Michele all'Adige (TN), Italy

* mmanica@fbk.eu

## Abstract

### Background

Emerging arboviral diseases in Europe pose a challenge due to difficulties in detecting and diagnosing cases during the initial circulation of the pathogen. Early outbreak detection enables public health authorities to take effective actions to reduce disease transmission. Quantification of the reporting delays of cases is vital to plan and assess surveillance and control strategies. Here, we provide estimates of reporting delays during an emerging arboviral outbreak and indications on how delays may have impacted onward transmission.

### Methodology/principal findings

Using descriptive statistics and Kaplan-Meyer curves we analyzed case reporting delays (the period between the date of symptom onset and the date of notification to the public health authorities) during the 2017 Italian chikungunya outbreak. We further investigated the effect of outbreak detection on reporting delays by means of a Cox proportional hazard model. We estimated that the overall median reporting delay was 15.5 days, but this was reduced to 8 days after the notification of the first case. Cases with symptom onset after outbreak detection had about a 3.5 times higher reporting rate, however only 3.6% were notified within 24h from symptom onset. Remarkably, we found that 45.9% of identified cases developed symptoms before the detection of the outbreak.

### Conclusions/significance

These results suggest that efforts should be undertaken to improve the early detection and identification of arboviral cases, as well as the management of vector species to mitigate the impact of long reporting delays.

**Data Availability Statement:** The data and code used in this article are available in Figshare, at https://dx.doi.org/10.6084/m9.figshare.23815071.

**Funding:** The epidemiological investigation and data collection were funded by the Directorate of Health and Social Welfare, Lazio Region, and the Local Public Health Units, Lazio Region. INMI acknowledges the financial support for research on emerging pathogens by the Italian Ministry of Health, grants to Ricerca Corrente linea 1. GG and GM acknowledge that this study was partially funded by EU grant agreement No 874850 MOOD and is catalogued as MOOD 074. MM, AdT and BC acknowledge that this research was supported by EU funding within the MUR PNRR Extended Partnership initiative on Emerging Infectious Diseases (Project no. PE00000007, INF-ACT). The contents of this publication are the sole responsibility of the authors and don't necessarily reflect the views of the European Commission. AdT and RR acknowledge funding from PRIN2020: Tackling mosquitoes in Italy: from citizen to bench and back, Pto. N. 2020XYBN88. BC acknowledges that this project has received from University of Sapienza funding within Bando Ricerca 2022-Progetti diRicercaMedin. N. RM12218148E7F756. The funders had no role in the study design, data collection and analysis, decision to publish, or preparation of the manuscript.

**Competing interests:** The authors have declared that no competing interests exist.

## Author summary

Timely detection of arboviral introduction and transmission is paramount to decrease the probability of further autochthonous transmission in non-endemic countries. When, despite all efforts, autochthonous transmission occurs, then an early detection of cases helps in bringing the outbreak under control. However, due to the still low frequency in Europe of local outbreaks of arboviruses transmitted by the vector mosquito species *Aedes albopictus*, little is known about the performance of public health surveillance in an ongoing emergency situation. This research investigates the reporting delays observed during the 2017 Italian chikungunya outbreak and provides estimate useful for modeling outbreak responses as well as suggesting the strengthening of the surveillance systems in the early detection and identification of arboviral cases.

## Introduction

Timely detection of infected cases enables public health authorities to take effective actions to reduce disease transmission in a population, limiting health and socio-economic impacts. In the case of mosquito-borne diseases, the longer an undetected viremic individual is left exposed to mosquito bites, the higher is the risk of subsequent transmission and of the spread of the pathogen. In the last decades, in European countries, autochthonous transmission of different emerging mosquito-borne diseases such as dengue and chikungunya occurred sporadically. Occasionally these caused local epidemics [1–3]. These events led several European countries and the European Centre for Disease Prevention and Control to develop surveillance systems and response guidelines for arboviral infections [4–6]. Estimates of the distribution of case reporting delays (the period between the date of symptom onset and the date of notification to the public health authorities) would represent a valuable indicator of the quality of the public health surveillance process and of health systems' preparedness [6], although other factors not related to public health efficiency play a role [7]. In fact, factors influencing the length of the reporting delay are expected to be highly heterogeneous across cases as they could be patient specific (e.g. time interval after symptom onset before seeking medical intervention), but they can also be influenced by physicians (e.g. time before visiting the patient or prescribing the proper diagnostic test), by the testing laboratories (e.g. time before sending the test result), and by the socio-technological infrastructure in place to report identified cases to the responsible public health authority.

Chikungunya virus (CHIKV) is a pathogen which belongs to the Togaviridae family (genus Alphavirus) and is transmitted by *Aedes* mosquito species to humans, that are a competent host [8]. CHIKV has continuously re-emerged in Africa and Asia during the last decade and has been routinely imported into Europe and America through travelers. CHIKV is a perfect example of the difficulties of early identification of pathogen circulation that public health authorities are increasingly facing in Europe [9]. One of the reasons is that a non-negligible fraction of human infections is asymptomatic and that symptomatic patients do not always seek medical care because of mild symptoms and/or lack of knowledge of *Aedes* transmitted diseases [10–12]. Additionally, general practitioners or clinicians may not promptly diagnose the disease because in non-endemic areas this could be confused with other clinical conditions (e.g. measles) due to low disease awareness [13] or more concerning disease (e.g. dengue) [3]. All these factors could contribute to an initially silent circulation of the pathogen in the human population, as occurred in the early phase of the two CHIKV outbreaks experienced in Italy in 2007 and 2017 [1]. In 2017, epidemiological investigations and active case findings identified a

total of 499 probable cases, suggesting three main foci (Anzio, Rome and Latina) within Lazio region [14] and an additional one in the Calabria region (Guardavalle) [15]. The outbreak in the Lazio region was identified when the National Reference Laboratory for Arboviral Infections at the National Institute of Public Health and the National Institute for Infectious Diseases analyzed serum and urine samples from a cluster of three suspected cases [16]. All cases resulted IgM positive on 6 and 7 September 2017 and were confirmed through a neutralisation test [14,16,17]. All cases showed symptoms (*i.e.* high fever above 38˚ C, severe joint pain and skin rash) but only one case was still symptomatic at the time of laboratory testing. None of the cases had an history of travel in endemic area in the past 15 days, they all lived in the same household and the onset of symptoms happened during their holiday in the coastal area of Anzio [14,16,17]. After the identification of the outbreaks, health authorities implemented blood safety restrictions (e.g.: suspension of blood collection) until 1 December 2017 [15,18] and reactive vector control measures by aerial spraying with pyrethroid insecticides, residual etofenprox-based insecticides and larvicide in street drains in both public areas and private houses [16,19]. The surveillance system in place in the Lazio region in 2017 included different surveillance tools: passive surveillance relying on medical practitioners reporting suspected cases, laboratory-based surveillance, and syndromic surveillance [20]. Based on national and regional guidelines, reporting of suspected arbovirus cases to local Public Health authorities is mandatory [14,17,21]. However, reporting time was not analyzed, despite its importance to inform preventive measures for future outbreaks.

In the present work, we analyze reporting delays during the 2017 chikungunya outbreak in the Lazio region (Italy) and discuss potential consequences in the control of the diseases spread.

## Materials and methods

We analyzed all suspected, probable, and confirmed chikungunya cases notified by Regional Service for Epidemiology, Surveillance and control of Infectious Diseases (SERESMI) of Lazio Region between 6[th] September 2017 and 5[th] February 2018 [14]. The criteria for suspected case definition were presence of symptoms (fever and joint pain) without history of travel to an endemic country in the 15 days preceding the symptom onset or living with or close to (<200m) a probable/confirmed case. The medical practitioner that suspects an arboviral infection in subjects who meet the clinical and epidemiological case definitions must report it to the competent regional Public Health authority in charge of the epidemiological surveillance (i.e. SERESMI) within 12 hours and collect samples to send for laboratory confirmation. A probable case was defined as any suspected case that tested positive for anti-CHIKV IgM on a single serum sample. A confirmed case was defined as any suspected case that tested positive for either CHIKV PCR or as a suspected case who tested positive for anti-CHIKV IgM on a single serum sample and whose positivity was either confirmed by sero-neutralization or sero-converted from negative to positive or by showing a fourfold increase of Ig titer in two subsequent samples taken at least 2 weeks apart (see [14] and S1 Text for more information on case definition). In the event of positive laboratory tests, the case must be immediately reported by the health authority to the administrative body of the Region and within 12 hours to the Ministry of Health and the National Institute of Health. The guidelines for reporting in effect at the time of the outbreak are included in S1 Text.

The dataset was anonymized, and the following information were provided for each case: date of symptom onset (if any), date of notification, presumed epidemiological link, and case definition (suspect, probable or laboratory confirmed). The date of symptom onset was obtained by interviews carried out during epidemiological investigations [14]. The date of

notification represents the date of reporting to SERESMI as a suspect case. This notification took place within 24-h from the assessment of compliance to the definition of suspect case [14]. The epidemiological link indicated the area within the Lazio region where the infection most likely occurred.

In this work, we defined the reporting delay associated with symptomatic cases as the number of days elapsed between the notification date and the date of symptom onset. In addition, we classified each case as occurred before outbreak detection (BOD) or after outbreak detection (AOD), by comparing the date of symptom onset with to the date of first notification (September 6, 2017 [17]). We investigated reporting delays using Kaplan-Meyer curves [22], by considering a dichotomous variable indicating whether the case was BOD or AOD. Given that information is available only for notified cases there is no censoring, and we could not consider undetected (i.e., asymptomatic, not seeking medical care) cases. Obtained findings were used to retrospectively evaluate how the public health response triggered by the outbreak detection affected notification timeliness. We further analyzed the effect of outbreak detection to reporting delays by means of a Cox proportional hazard model [22]. We ran two sensitivity analyses: in the first, we excluded all cases with a reporting delay longer than 30 days to minimize potential recall bias in the identification of the date of symptom onset, in the second we considered only confirmed cases.

Finally, we investigated the proportion of cases that were notified before the end of their infectious period and, under the assumption that notification would prevent further transmission (*ie* self-protection from bites and implementation of measures for vector control), we estimated the consequent avoided number of days at population level during which uncontrolled onward transmission can occur. We assigned the start and length of the infectious period following the viremia time profile as in Cauchemez et al, 2014 [23]. For each case, the first day of the infectious was sampled one or two days (with equal probability 0.5) before the date of symptom onset, while the length of the infectious period was uniformly sampled between 5 to 8 days. We repeated the process 1000 times to obtain a distribution of infectious period for each case identified during the outbreak. Then, we computed the percentage of cases that were notified before the end of their infectious period and the percentage of days of uncontrolled onward transmission that may have been avoided at population level. In addition, we carried out a sensitivity analysis where we estimated the infectious period of chikungunya cases by fitting a truncated Poisson distribution to data on the duration (in days) of fever in identified imported chikungunya cases in Italy [24], under the assumption that the infectious period overlaps the manifestations of disease symptoms [8]. Then, as in the main analysis, we simulated 1000 infectious period for each case using the estimated truncated Poisson distribution.

## Results

In total, 414 cases were notified in the Lazio region, of which 312 with epidemiological link to Anzio, 73 to Rome, 8 to Latina and 21 scattered across the region (Fig 1). Overall, 200, 202, and 12 cases were classified as confirmed, probable and suspected, respectively. Two cases (one in Anzio and one in Rome) with missing information regarding their symptoms' onset were excluded from the proposed analysis. The epidemiological investigation pointed out that 189 (45.9%) patients developed symptoms before outbreak detection (BOD); 223 (54.1%) afterward (AOD). The median reporting delay was 15.5 overall (interquartile range 7–36.25), 33 days BOD (interquartile range 19–52) and 8 days AOD (interquartile range 5–15).

The earliest symptom onset retrospectively identified through epidemiological investigation was on 26th June in Anzio, 72 days before the outbreak detection on September 6th (Fig 1). In Rome and Latina, the other two foci, it was on 20th (17 days BOD) and 13th (24 days BOD)

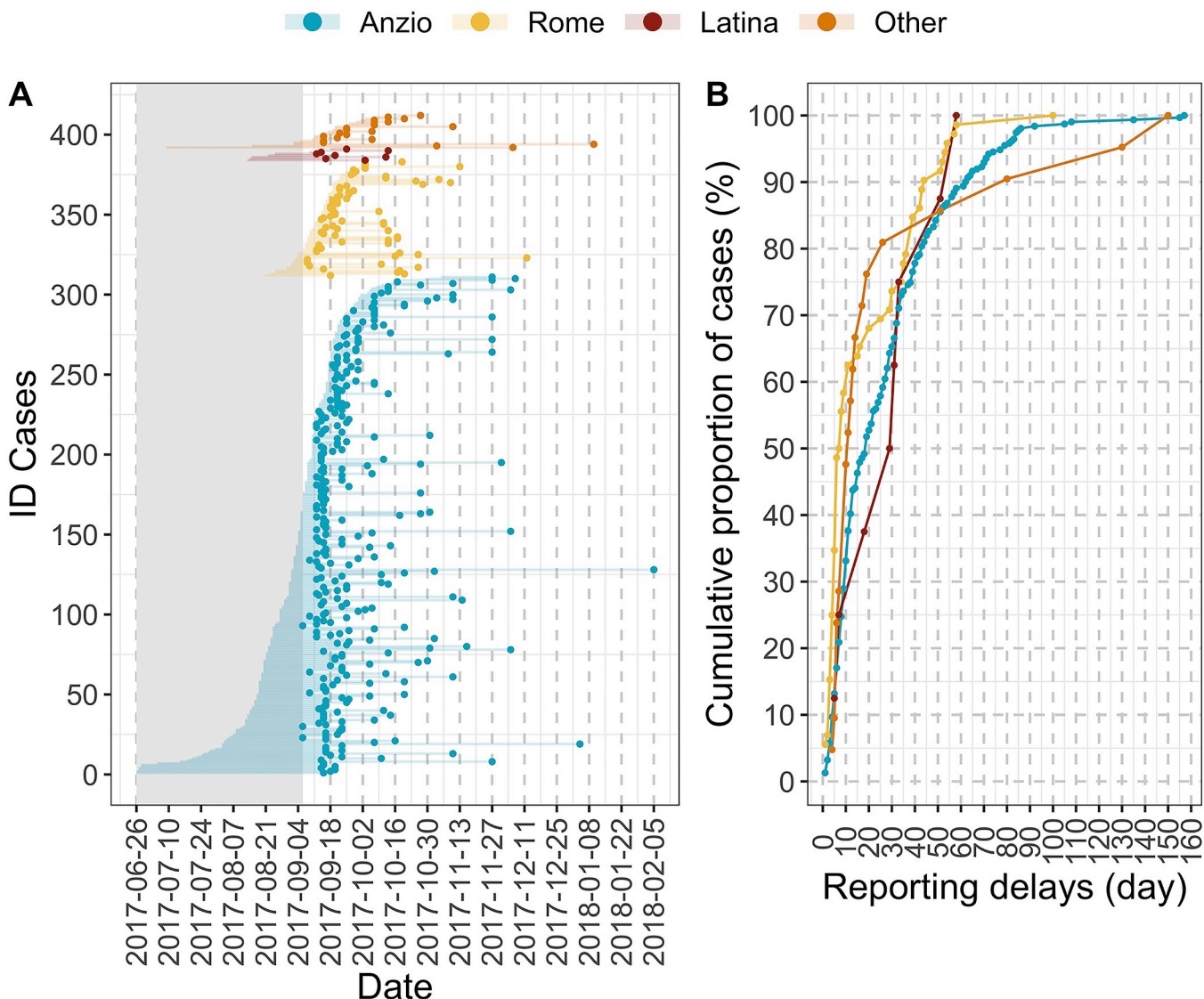

**Fig 1.** A) Time of key events associated with identified cases. On the x-axis the date, on the y-axis the cases ID. Points represent the time at notification. Lines represent the cases' reporting delay. Different colors are used to group cases into the different epidemiological foci. The grey area defines the period before the detection of the outbreak. B) Cumulative proportion of cases by reporting delays and foci. On the x-axis the reporting delay, on the y-axis the cumulative proportion of cases. Different colors are used to group cases into the different epidemiological foci.

August, respectively. The symptom onset for the first notified cases (earliest notifications, all linked to the Anzio cluster) happened on 5th, 11th, and 25th of August (32, 26, 12 days BOD). On the other hand, the earliest symptom onset for the first notified cases in Rome (8th September) and Latina (12th September) was on 2nd September and 25th August, respectively. The percentage of notified cases within 24h from symptom onset was 3.6% AOD. In the first week after the outbreak detection, reporting involved both recent and retrospective cases (Figs 1 and 2), while after 4 weeks (ie after epidemiological week 40) the notification increasingly involved less recent cases (ie with a greater reporting delay, as shown in Fig 2, panel D).

Using Kaplan-Meier curves, we estimated a median reporting delay of 33 days (95%CI: 31–37) BOD and 8 days (95%CI: 7–9) AOD (Fig 3). Similarly, the Cox model estimated a hazard ratio of being notified of 3.45 (95%CI: 2.80–4.25) AOD compared to BOD; in other words,

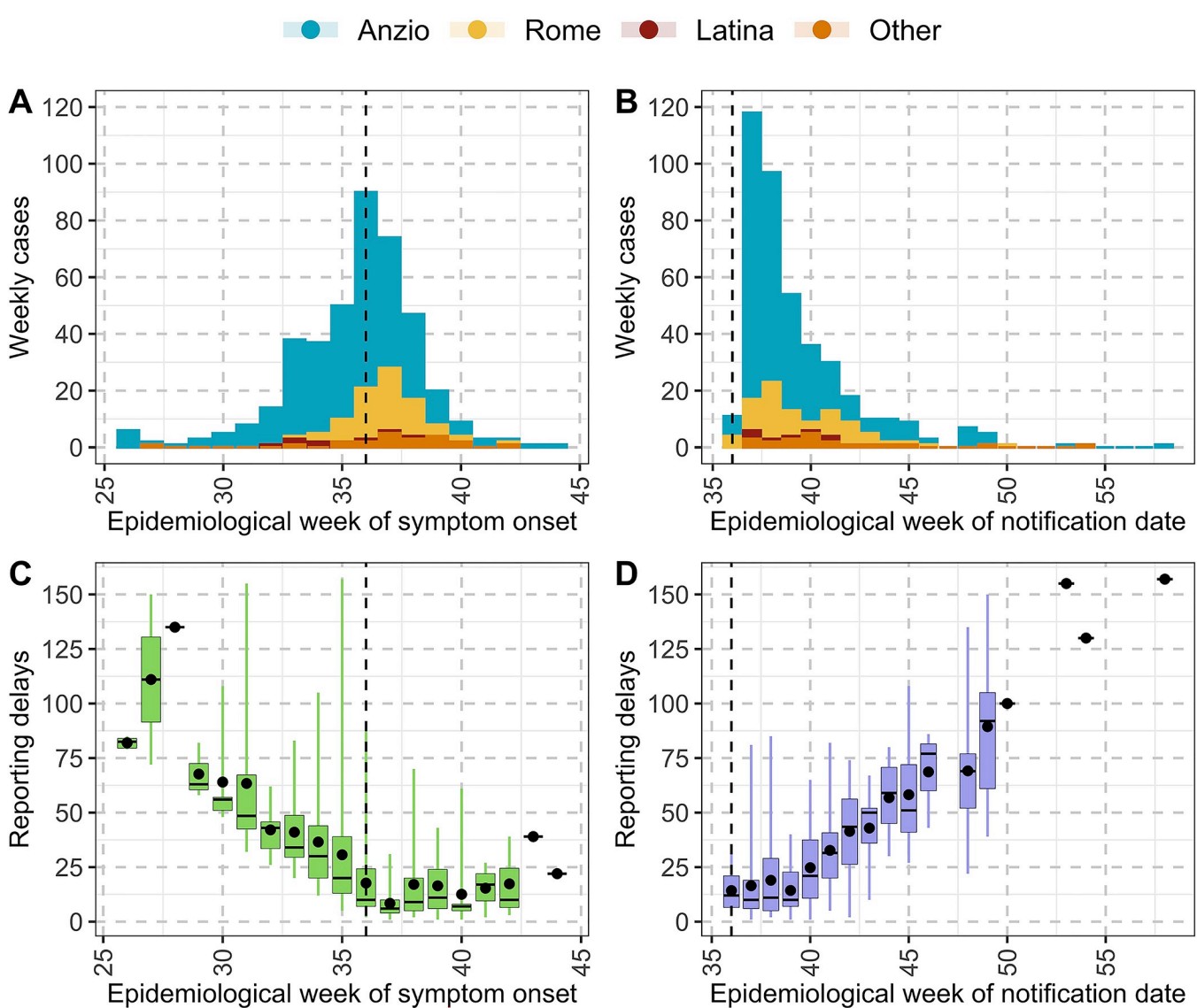

**Fig 2.** A) Weekly number of identified cases by date of symptom onset. B) As A) but by date of notification. Different colors are used to group cases into the different epidemiological foci. C) Distribution of reporting delay by week of symptom onset; boxplots represent median values (horizontal lines), interquartile ranges (box limits), minimum and maximum values (vertical lines) and points represent mean values; counting of epidemiological weeks was extended to the following year (2018). D) As C) but by week of notification. Dashed vertical lines represent the week of outbreak detection.

among all cases that will be eventually identified, the rate of notification was about 3.5 time higher in cases with symptoms' onset AOD compared to cases with date of symptom onset BOD.

Results of sensitivity analyses confirmed the decrease in the length of reporting delay AOD and showed a lower reporting delay mainly due to the exclusion of patients with a reporting delay over 30 days. Among the 277 records obtained excluding the records with a reporting delay > 30 days (to avoid recall bias), 196 (70.8%) of which concerned cases with date of symptom onset AOD. The median reporting delay was 9 days (interquartile range: 6–16), 18 (interquartile range 11–25) and 7 (interquartile range: 4.75–11) overall, BOD and AOD, respectively. Results of Kaplan-Meier curve estimated a median reporting delay of 18 days (95%CI: 13–29) BOD and 7 days (95%CI: 6–8) AOD confirming the reduced time to

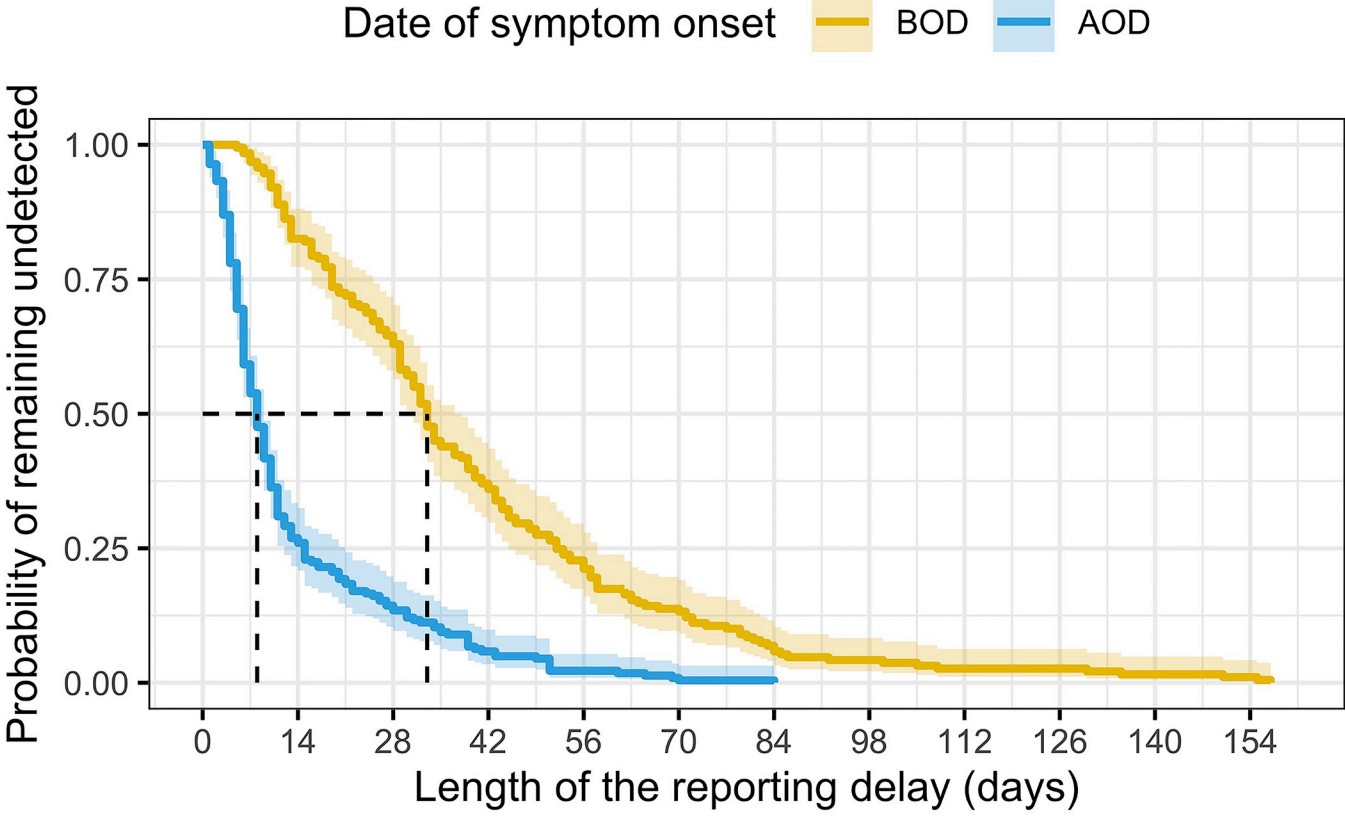

**Fig 3. Kaplan Meier curve.** Probability of the length of the reporting delay after symptom onset. On the x-axis the length of the reporting delay, on the y-axis the probability of remaining undetected. The table below represent the cumulative number of notifications at the corresponding time point (x-axis above).

notification after outbreak detection. The Cox model estimated a hazard ratio of being notified of 2.6 (95%CI: 2.0–3.4) AOD compared to BOD. Considering only confirmed cases resulted in 199 valid record, 156 (78.4%) of which concerned cases with date of symptom onset AOD. The median reporting delay was 8 days (interquartile range: 5–11.5), 13 (interquartile range 10.5–29) and 6 (interquartile range: 4–10) overall, BOD and AOD, respectively. Results of Kaplan-Meier curve estimated a median timeliness of 13 days (95%CI: 12–26) BOD and 6 days (95%CI: 6–8) AOD confirming the reduced reporting delays after outbreak detection. The Cox model estimated a hazard ratio of being notified of 2.9 (95%CI: 2.1–4.2) AOD compared to BOD.

Following [23], we assumed the infectious period to start one or two days before symptom onset and last between 5 to 8 days. We found that, on average 17% (95%CI: 15.3–18.9%) of cases had a notification time shorter than their infectious period. This percentage was almost zero BOD, 0.9% (95%CI: 0.– 2.1%) but increased up to 30.7% (95%CI: 27.4–34.1%) after outbreak detection (AOD). We estimated no reduction in the number of days per person when onward transmission can occur BOD while it was 0.3 days per person AOD (95%CI: 0.2–0.3) and 0.28 days per person on the whole period (95%CI: 0.2–0.3), respectively. This suggests an

average percentage reduction in the number of infectious days at population level of 0.1% AOD (95%CI: 0–0.2%), 8% BOD (95%CI: 6.9–9.1%) and 4.4% in the whole period (95%CI: 3.7–5%). A sensitivity analysis assuming no pre-symptomatic infectiousness and a length of the infectious period similar to the symptomatic period provided comparable results, see supplementary S1 Text.

## Discussion

The delay between symptom onset and notification to the public health system is a critical parameter in controlling outbreaks [25]. Such delays have been previously suggested to be one of the key drivers of the size and spread of the 2017 chikungunya outbreak in the Lazio region, Italy [15]. The analysis of reporting delays conducted in this study showed that even after detection of the outbreak by public health authorities, the length of these delays (median 8 days) was still comparable to the duration of the infectious period (5 to 8 days [23]). Since case-based interventions (such as insecticide spraying around the residence of a case) can only occur after case detection, these delays significantly impair the ability of control measures in disrupting pathogen transmission. Therefore, the resulting percentage reduction of days when onward transmission can occur at population level was not high (8% on average) even under the assumption that notification of a case is enough to prevent further transmission.

The reporting delay for the case with the earliest symptom onset (72 days) is higher than the range previously observed for the notification of imported cases in Italy (from 2 to 58 days, 18.6 days on average) [26,27]. Data showed that reporting delays greatly improved after the detection of the first cases, with a 3.5 higher rate of reporting and a mean reporting delay of 13.3 days after outbreak detection compared to 39.9 days before outbreak detection. Nonetheless, it is likely that a large proportion of patients, who presented no or mild symptoms, remained undetected by the public health systems. This hypothesis is supported by evidence gathered after the Italian 2007 outbreak when a seroprevalence study revealed that symptomatic individuals correctly identified by active surveillance represented only 63% of the total number of cases [28]. Indeed, the degree and share of transmission episodes originating from asymptomatic individuals may affect the effectiveness of an outbreak response based on vector control around the place of residence of notified cases.

Reduction of reporting delays should be a priority as suggested by previous theoretical assessment of public health strategies to reduce the risk of autochthonous transmission [29,30]. However, a recent modeling study [31] estimated that shortening the delay from symptom onset to intervention (from an average of 6 to an average of 3 days) would have brought limited reductions in the size of the 2017 chikungunya outbreak. Underlying reasons were the limited effectiveness of control interventions and the potentially wider distribution of transmission distances compared to the area targeted by reactive vector control interventions. However, the same study suggested that an earlier outbreak detection by just two generations of transmission (about 3 weeks) could have reduced the number of cases occurring after outbreak detection by over two thirds, therefore stressing the importance of a surveillance system capable of quickly detecting the introduction of tropical arboviruses, as also observed in France where anticipating vector control intervention by 10 days would have reduced the final outbreak size by 30–40% fewer cases [25,32]. This may be achieved, at least for chikungunya and other non-endemic diseases, by means of preventive rather than reactive measures such as strengthening surveillance of imported cases, raising awareness in both travelers and general practitioners [13,33] and an integrated management of vector population [31,34].

It is relevant to highlight that this study has some limitations. First, all estimates are computed only from the analysis of notified cases, due to unavailable data on the total infected

population. Likely, this has biased our results toward shorter reporting delays and an overestimation of the reduction in the probability that onward transmission can occur. In addition, the lack of information on the individual dates of infection and duration of infectiousness may have also led to an overestimation of the reduction in the number of infectious days at population level. Second, data on symptom onset was obtained for each case by interviews carried out during epidemiological investigations and longer reporting delays may correlate with an increased bias in the identification of the exact date of symptom onset. Third, the usefulness of the provided estimates is limited to the description of the 2017 Italian outbreak and might not apply to other settings or conditions. Nevertheless, these are the first estimates of reporting delays during a large exotic arbovirus outbreak in a temperate region.

The frequency of autochthonous transmission events of chikungunya and related arboviruses in Europe is likely to increase in the future due to the growing importation into European countries of infected travelers [35,36] and the progressive expansion of mosquito populations (e.g. *Ae. albopictus)* representing competent vectors for CHIKV transmission [32].

Early detection of imported chikungunya cases as well as the initial autochthonous transmission events is of the utmost importance to contain or even prevent the spread and size of any future chikungunya outbreak [23,31]. The aim of improving detection and notification timeliness can be achieved only by a sustained and coordinated effort involving many activities. For instance, it is essential not only to raise awareness on chikungunya infection in the general population and in individuals travelling to and from endemic countries but also provide information and training on diagnosis and reporting for medical practitioners. Moreover, medical practitioners should be supported by developing improved diagnostic algorithms as well as improving the information flow by minimizing the time delay between any response steps in the public health surveillance process. An important surveillance tool could be provided by syndromic surveillance where clearly defined sets of symptoms could be used to flag suspected transmission events. Also, novel data sources such as search generated data could be potentially integrated in the surveillance system in order to enhance the early detection of viral transmission. Another critical aspect in reducing reporting delay involves speeding up laboratory procedure both in national reference laboratories as well as in first-line responding hospital laboratories. This can be achieved by strengthening laboratory capability to face increased workload due to increased testing and by increasing laboratory detection capacity challenged by the paucity of available commercial kits, the cross-reactivity between related arboviruses affecting serology testing and the short viremia period. Finally, routine evaluation of reporting performances should be planned and routinely carried out to ensure the surveillance system preparedness and responsiveness.

In conclusion, our results strongly suggest that public health authorities should take actions to increase travelers' awareness to potential health threats, set up early detection surveillance systems and plan the surveillance and management of the local vector population [37,38] to reduce the risk of autochthonous transmission and local outbreaks.

## Supporting information

**S1 Text. Supplementary information on the reporting system, control interventions and sensitivity analysis**
(PDF)

## Author Contributions

**Conceptualization:** Mattia Manica, Angelo Solimini, Roberto Rosà, Beniamino Caputo.

**Data curation:** Mattia Manica, Francesco Vairo.

**Formal analysis:** Mattia Manica.

**Funding acquisition:** Stefano Merler, Roberto Rosà.

**Investigation:** Francesco Vairo.

**Methodology:** Mattia Manica, Giovanni Marini, Angelo Solimini, Giorgio Guzzetta, Piero Poletti, Beniamino Caputo.

**Resources:** Paola Scognamiglio, Francesco Vairo.

**Software:** Mattia Manica.

**Supervision:** Paola Scognamiglio, Alessandra della Torre, Stefano Merler, Roberto Rosà.

**Validation:** Giovanni Marini, Chiara Virgillito.

**Visualization:** Mattia Manica.

**Writing – original draft:** Mattia Manica.

**Writing – review & editing:** Giovanni Marini, Angelo Solimini, Giorgio Guzzetta, Piero Poletti, Paola Scognamiglio, Chiara Virgillito, Alessandra della Torre, Stefano Merler, Roberto Rosà, Francesco Vairo, Beniamino Caputo.

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
