## [Decision Letter · Decision Letter 0]

20 Jun 2023

Dear Dr. Manica,

Thank you very much for submitting your manuscript "Reporting delays of chikungunya cases during the 2017 outbreak in Lazio region, Italy." for consideration at PLOS Neglected Tropical Diseases. As with all papers reviewed by the journal, your manuscript was reviewed by members of the editorial board and by several independent reviewers. In light of the reviews (below this email), we would like to invite the resubmission of a significantly-revised version that takes into account the reviewers' comments. 

We cannot make any decision about publication until we have seen the revised manuscript and your response to the reviewers' comments. Your revised manuscript is also likely to be sent to reviewers for further evaluation.

Sincerely,

Ran Wang, M.D.

Academic Editor

Justin Remais

Section Editor

Reviewer's Responses to Questions

**Key Review Criteria Required for Acceptance?**

**Methods**

-Are the objectives of the study clearly articulated with a clear testable hypothesis stated?

-Is the study design appropriate to address the stated objectives?

-Is the population clearly described and appropriate for the hypothesis being tested?

-Is the sample size sufficient to ensure adequate power to address the hypothesis being tested?

-Were correct statistical analysis used to support conclusions?

-Are there concerns about ethical or regulatory requirements being met?

Reviewer #1: The methods are appropriate for the analyses conducted. Specific clarifications are noted:

1. To help understand the presented information on timing of infection reporting relative to outbreak detection it would be useful for readers to understand some of the details of the reporting system or systems used in different regions of Italy. Is chikungunya infection reporting mandatory? Please describe the SERESMI reporting system so persons outside Italy can better appreciate the context. Who reports the case information and what are the details of the information which are requested? 

2. On page 11, the authors state that notification of infection can lead to a reduction in infectious days. Could the authors clarify what is meant here? Is this the cumulative infectious because of the presumed impact of case recognition on the overall number of cases infected? For each infected individual there would be no change/reduction in the infectious days, but at the population level there could be. 

3. The assumption that the symptomatic period and infectious period overlap may not be correct. In most situations the infectious period would precede the infectious period and then seroconversion and reduced infectiousness would be evident during the most symptomatic phase of infection.

Reviewer #2: The study has a clearly articulated hypothesis with subsequent analysis that adequately tests that hypothesis. The population studies is clearly described, and the samples used in analysis are aa comprehensive ans inclusive as possible given reporting limitations. The paper in its current form sufficiently answers the research question.

Reviewer #3: 1. Reporting delay can be the result of patients delaying their presenting for care, especially when the symptoms are mild, or delay in sending the specimens for testing or delay in the laboratory sending the results back to the patient and the health provider. It is also not known how the results gets reported to the surveillance system. It would be useful if the authors can briefly describe how the outbreak emerged, how the cases are detected (from the surveillance system or from clinicians in the community or both) and what is involved in cases processing from suspect cases to probable and confirmed cases (was testing done at surveillance sites or were specimens transported to laboratories?) and how the cases are reported to the surveillance system? 

2. What were the criteria for establishing Sept 6 2017 as the “date of first notification”? was it based on a cluster of confirmed cases? What measures were put into place once an outbreak is officially declared? Were family members of cases tested for additional case finding? 

3. Vector Control: Since there is no treatment or vaccines for Chikungunya, prevention and control interventions generally rely on case-based interventions (mosquito spraying and other vector control measures around the residence of cases)., the proportion of asymptomatic but infectious individuals in a community would have a varying impact on the effectiveness of the outbreak response. Were community-wide spraying of public spaces part of the vector control in addition to interventions around the place of residence of notified cases?

**Results**

-Does the analysis presented match the analysis plan?

-Are the results clearly and completely presented?

-Are the figures (Tables, Images) of sufficient quality for clarity?

Reviewer #1: The results align with the planned analysis and no deficiencies are noted. Figures are appropriate. Supplemental figures are not included and could not be reviewed.

Reviewer #2: Results are completely presented and figures clear and aesthetically pleasing.

Reviewer #3: The results are clearly presented and the figures are of sufficient quality for clarity.

**Conclusions**

-Are the conclusions supported by the data presented?

-Are the limitations of analysis clearly described?

-Do the authors discuss how these data can be helpful to advance our understanding of the topic under study?

-Is public health relevance addressed?

Reviewer #1: The conclusions are appropriate and supported by the data. The most relevant limitations are clearly stated. 

The primary deficiency is that specific recommendations on how to improve the process or timeliness of infection reporting in Italy are not provided. This additional specificity would improve the practical relevance of the analysis and findings.

Reviewer #2: The conclusions are supported by the results and the authors provide helpful context that help the reader appreciate the public health significance of the findings.

Reviewer #3: The discussion is good but it is not clear how the authors presented the quantitation of the health burden. what is the definition of health burden and how was that quantitated? 

What are the limitations of their study - did they provide readers with a better understanding of reporting delays and how that impacted the final health burden of this outbreak compared to previous studies? 

Line 275: What is meant by,” …..and the short radius of treatment compared to the potential distribution of transmission distances”? treatment? Please explain.

**Editorial and Data Presentation Modifications?**

Reviewer #1: Page 5, Line 147 – dichotomous rather than dichomotic 

Page 6, Line 161 – truncated rather than truncate

Page 11, Line 236 – ‘the symptomatic period’ rather than symptom

Reviewer #2: Line 76 - chikungunya should be decapitalized for consistency internally and with the existing literature

Line 153 - analysis should read analyses

Line 180 - figure caption shoukd state “events” rather than “evens”

Reviewer #3: (No Response)

**Summary and General Comments**

Reviewer #1: The supplemental materials do not appear available to review and so were not reviewed.

Reviewer #2: This study answers an important epidemiological question using rigorous methods. The results are presented in a clear manner, elucidating important findings without overstating. The paper provides helpful insight that can help the scientific community better understand the ramifications of public health campagins during disease outbreaks.

Reviewer #3: This manuscript describes a study to quantify reporting delays of cases and assess the impact of these delays on the final health burden during an outbreak of Chikungunya in Italy. Reporting delays of infectious disease outbreaks have major impact on the development and implementation of disease prevention and control strategies. Hence this type of study should be a high priority during inter-epidemic periods to inform how surveillance and health systems should be optimised for managing future outbreaks. The aims are clearly stated and the methods and results are well described. However, there is insufficient information on aspects of the outbreak and the control strategies for readersts to interpret the results and draw conclusions.
---

## [Decision Letter · Decision Letter 1]

22 Aug 2023

Dear Dr. Manica,

We are pleased to inform you that your manuscript 'Reporting delays of chikungunya cases during the 2017 outbreak in Lazio region, Italy.' has been provisionally accepted for publication in PLOS Neglected Tropical Diseases.

Best regards,

Ran Wang, M.D.

Academic Editor

Justin Remais

Section Editor

Reviewer's Responses to Questions

**Key Review Criteria Required for Acceptance?**

**Methods**

-Are the objectives of the study clearly articulated with a clear testable hypothesis stated?

-Is the study design appropriate to address the stated objectives?

-Is the population clearly described and appropriate for the hypothesis being tested?

-Is the sample size sufficient to ensure adequate power to address the hypothesis being tested?

-Were correct statistical analysis used to support conclusions?

-Are there concerns about ethical or regulatory requirements being met?

Reviewer #1: The authors have responded to the comments of the reviewers and have improved the reporting of the methods.

Reviewer #3: The revised method section is now much clearer.

**Results**

-Does the analysis presented match the analysis plan?

-Are the results clearly and completely presented?

-Are the figures (Tables, Images) of sufficient quality for clarity?

Reviewer #1: Yes, the results match the analysis.

Reviewer #3: The results now make more sense as the authors have provided more information on the how the surveillance and notification system works.

**Conclusions**

-Are the conclusions supported by the data presented?

-Are the limitations of analysis clearly described?

-Do the authors discuss how these data can be helpful to advance our understanding of the topic under study?

-Is public health relevance addressed?

Reviewer #1: The conclusions are appropriate for the data.

Reviewer #3: The conclusions are supported by the data presented and the limitations of the study have been described. The findings of this study have important public health implcations and the authors have made that clear.

**Editorial and Data Presentation Modifications?**

Reviewer #1: I have not reviewed this version of the manuscript for minor revisions. Those indicated previously have been addressed by the authors.

Reviewer #3: none required.

**Summary and General Comments**

Reviewer #1: The manuscript is a valuable contribution to the literature.

Reviewer #3: The authors have addressed all the reviewers' concerns. The methods, results and discussion sections are now much clearer and more informative. I recommend this paper for publication.

---

## [Editor Report · Acceptance letter]

31 Aug 2023

Dear Dr. Manica,

We are delighted to inform you that your manuscript, "Reporting delays of chikungunya cases during the 2017 outbreak in Lazio region, Italy.," has been formally accepted for publication in PLOS Neglected Tropical Diseases.

Best regards,

Shaden Kamhawi

co-Editor-in-Chief

Paul Brindley

co-Editor-in-Chief
